# Advanced design and Engi-economical evaluation of an automatic sugarcane seed cutting machine based RGB color sensor

**Abdallah Elshawadfy Elwakeel** [1]*, **Loai S. Nasrat**[2], **Mohamed Elshahat Badawy**[3], **I. M. Elzein** [4], **Mohamed Metwally Mahmoud**[5]*, **Kitmo** [6]*, **Mahmoud M. Hussein**[5,7], **Hany S. Hussein**[5], **Tamer M. El-Messery**[8], **Claude Nyambe**[8], **Salah Elsayed**[9], **Manar A. Ourapi**[1]

1 Agricultural Engineering Department, Faculty of Agriculture and Natural Resources, Aswan University, Aswan, Egypt, 2 Electrical Power Engineering Department, Faculty of Engineering, Aswan University, Aswan, Egypt, 3 Agricultural Engineering Research Institute - Dokki – Giza, Egypt, 4 Department of Electrical Engineering, College of Engineering and Technology, University of Doha for Science and Technology, Doha, Qatar, 5 Electrical Engineering Department, Faculty of Energy Engineering, Aswan University, Aswan, Egypt, 6 University of Maroua, National Advanced School of Engineering of Maroua, Department of Renewable Energy, Maroua, Cameroon, 7 Department of Communications Technology Engineering, Technical College, Imam Ja'afar Al-Sadiq University, Baghdad, Iraq, 8 International Research Centre "Biotechnologies of the Third Millennium", Faculty of Biotechnologies (BioTech), ITMO University, St. Petersburg, Russia, 9 Agricultural Engineering, Evaluation of Natural Resources Department, Environmental Studies and Research Institute, University of Sadat City, Sadat City, Egypt

* Abdallah_elshawadfy@agr.aswu.edu.eg, Manarourpi@gmail.com (AEE); Metwally_M@aswu.edu.eg (MMM); kitmobahn@gmail.com (K)

**Data Availability Statement:** All relevant data are within the manuscript.

## Abstract

There are many problems related to the use of machine learning and machine vision technology on a commercial scale for cutting sugarcane seeds. These obstacles are related to complex systems and the way the farmers operate them, the possibility of damage to the buds during the cleaning process, and the high cost of such technology. In order to address these issues, a set of RGB color sensors was used to develop an automated sugarcane seed cutting machine (ASSCM) capable of identifying the buds that had been manually marked with a unique color and then cutting them mechanically, and the sugarcane seed exit chute was provided with a sugarcane seed monitoring unit. The machine's performance was evaluated by measuring the damage index at sugarcane stalk diameters of 2.03, 2.72, 3.42, and 3.94 cm. where two different types of rotary saw knives had the same diameter of 7.0 in/180 mm the two knives had 30 and 80 teeth, also we used five cutting times of 1000, 1500, 2000, 2500, and 3000 ms. All tests were done at a fixed cutting speed of 12000 rpm. In addition, the machine's performance was evaluated by conducting an economic analysis. The obtained results showed that the most damage index values were less than 0.00 for all cutting times and sugarcane stalk diameters under testing, while the DI values were equal zero (partial damage) for sugarcane stalk diameter of 3.42 cm at cutting times of 2000 ms and 2500 ms, in addition to the DI values being equal zero (extreme damage) for sugarcane stalk diameter of 3.94 cm at cutting times of 1500 ms and 2000 ms. The economic analysis showed that the total cost of sugarcane seeds per hectare is 70.865 USD. In addition, the ASSCM can pay for itself in a short period of time. The payback time is 0.536 years, which means that the ASSCM will save enough money to pay for itself in about 6.43 months.

**Funding:** The authors extend their appreciation to the Deanship of Scientific Research at King Khalid University for funding this work through General Research Project under grant number (RGP.2/125/45).

**Competing interests:** The authors have declared that no competing interests exist.

**Abbreviations:** *ASSCM*, automatic sugarcane seed cutting machine; $C_a$, The annualized investment cost; $C_{ac}$, The annualized capital cost; $C_{cc}$, The total capital cost; $C_{dp}$, The cost of fresh sugarcane stalks per hectare; $C_{fd}$, The cost of fresh sugarcane stalks per kg; $C_m$, The maintenance costs; $C_s$, The sugarcane seed cutting cost per hectare; *D*, The numbers of days the ASSCM operates in a year; *DC*, Direct current; *ED*, Extreme damage; *HSV*, Herpes simplex virus; *i*, The inflation rate; *ID*, Damage index; $M_d$, The amount of sugarcane seed cutting per day; ms, Millisecond; $M_y$, The numbers of hectares can be planted per year; n, The payback time; $n_{ED}$, number of sugarcane seeds with extreme damage; $n_{PD}$, number of sugarcane seeds with partial damage; $n_{SD}$, number of sugarcane seeds without damage; *PD*, Partial damage; $P_{ED}$, Weight attributed of extreme damage; $P_{PD}$, Weight attributed of partial damage; $P_{SD}$, Weight attributed of without damage; *PV*, Photovoltaic; $S_1$, The saving after the first year; $S_d$, The saving obtained from the ASSCM per day; $S_d$, The saving obtained per day; *SD*, Without damage; $S_{ha}$, Savings obtained per hectare of sugarcane seeds; $SP_c$, The selling price of sugarcane seeds per hectare; $V_a$, The salvage value.

Finally, we suggest using a rotary saw knife with 80 teeth and a cutting time of 2000 ms to cut sugarcane stacks with an average diameter of 2.72 cm. This will result in higher performance and lower operating costs for the ASSCM.

# 1. Introduction

Sugarcane is a crucial cash crop in the industrial sector, responsible for about 80% of global sugar production. Additionally, it is very valuable as a biofuel and renewable energy source [1–3]. In 2023, the global sugarcane production reached an estimated 187.88 million metric tons, with Egypt contributing 2.93 million metric tons [4]. Sugarcane is a plant that reproduces by asexual means and has a cylindrical shape with an upright, branching, clustered, and drooping stem. Since sugarcane nodes include buds, it is necessary to carefully choose suitable sugarcane nodes for planting in agricultural production. Real-time cutting is a method employed in conventional sugarcane cultivation that involves inserting the entire sugarcane stalk into the planting trench, dividing it into multiple segments, each measuring approximately 30–40 cm, ensuring that each segment contains at least two buds, and subsequently covering each segment completely with soil manually. With the exception of mechanical plowing and road transportation, almost all other tasks are carried out manually, necessitating substantial human and material resources for seed storage and selection. Automated production may significantly enhance the efficiency of sugarcane seed selection and seed value [5].

The majority of planting machines around the world are missing the function of bud protection in the automatic cutting process of sugarcane seeds. A real-time automatic planter is often a fixed-length cutter with higher efficiency than artificial ploughing [1–3, 6, 7]. Nevertheless, the rate of bud injury is extremely high. The pre-cutting planter is more efficient than the real-time cutting machine because it uses pre-cut seeds to minimize seed bud damage [8–10]. Unfortunately, the cutting efficiency of the sugarcane cutting machine is low, as it is unable to automatically locate the sugarcane bud [11, 12]. For that, it is necessary to develop a type of seed cutting technology and equipment that avoids harming the seed buds to increase the efficiency of seed cutting and the quality of sugarcane seed. Identification and location of sugarcane nodes require further non-destructive examinations. For products from agriculture, there are currently a wide range of non-destructive testing techniques available [5], some developed based on machine vision and deep learning technology to recognize sugarcane nodes and prevent their damage when slicing sugarcane seeds [13–15]. For instance, a feature vector that measures the gradient of the columns was created to calculate the gradient value. Additionally, a method for finding the location was designed to locate the nodes of the sugarcane. The average recognition rate achieved was 93.00% [16]. However, the accuracy of identifying sugarcane with slight variations between the node and internode was just 17.00%. Moshashai et al. [13] conducted a preliminary study on the detection of sugarcane nodes using a threshold segmentation technique for a grayscale picture. Lu et al. [17] used a threshold to partition the color space of a sugarcane picture into segments representing the herpes simplex virus (HSV). They then generated a composite image by combining the inverse images of the H and S components. The composite image's blocks were classed and recognized using a support vector machine, resulting in an average identification rate of 93.36% for stem nodes. Huang et al. [18] developed a rectangular template that traveled horizontally over the sugarcane picture with a defined step length. They then calculated the average gray value on the G-B component image of the sugarcane. The location of the stem nodes was estimated using the maximum average

gray value, resulting in a recognition rate of 90.77%. However, the accuracy of the estimation was influenced by the step length and template width. Chen et al. [19] proposed a method for identifying sugarcane stem nodes based on the extreme point of a vertical projection function, with a recognition rate of 95.0% for three stem nodes. Zhou et al. [16] proposed a new machine vision-based sugarcane seed cutting technology. The sugarcane seed cutting system consists of mechanical, electrical, and optical processing components. Sugarcane stems have a 93% identification rate offline. Many neural network techniques have been successfully applied to the recognition of agricultural goods due to the rapid growth of deep learning [5, 17, 20–23]. Neural network recognition methods use both two-stage and one-stage procedures. In order to enhance the precision of recognition, it is common practice to use two-stage neural networks for the identification of agricultural goods [19, 24–26]. Li et al. [27] used an enhanced YOLOv3 network to increase the accuracy of sugarcane stem node recognition to 90.38%.

The application of machine vision in sugarcane seed cutting is currently constrained by a number of issues, as noted by [16, 28, 29]: 1. The cost of constructing and operating the sugarcane plantation is high, resulting in expensive seed cuttings or per acre expenditures. 2. It is necessary to remove the leaves from the sugarcane plant in order to expose just the stem. Performing this action improperly might result in extra expenses and even harm to the buds. 3. The difficulties are caused by the machine's slow speed, poor real-time performance, low identification efficiency, and high cost. 4. Additionally, the machine is unable to distinguish between a healthy bud and a damaged or wounded one.

The aim of the current study is to design an ASSCM using RGB color sensors and Internet of Things technology (IoT) to address the challenges faced in using machine vision for cutting sugarcane seeds. The proposed system will be tested with different rotary knives of (30 teeth and 80 teeth), different cutting times of (1000, 1500, 2000, 2500, and 3000 ms) and different sugarcane stack diameters of (2.03, 2.72, 3.42, and 3.94 cm) to estimate the damage index and damage frequency. Finally, the economic features of the ASSCM will be studied. The utilization of the IoT and RGB color sensors can aid in the creation of an ASSCM that uses inexpensive local technology, thereby significantly reducing operating costs. The machine is easy to operate and does not require advanced programming skills and technology, making it cost-effective. During the cutting process of the ASSCM, the sugarcane seeds (buds) that need to be cut can be manually identified and colored to avoid cleaning the sugarcane stalks before cutting them and prevent cutting infected cuttings and damaging of buds.

## 2. Materials and methods

### 2.1. Principle of operation

The sugarcane stalk after cleaning consists of a node area, an internode area, a leaf scar, and a sugarcane bud, as shown in Fig 1.

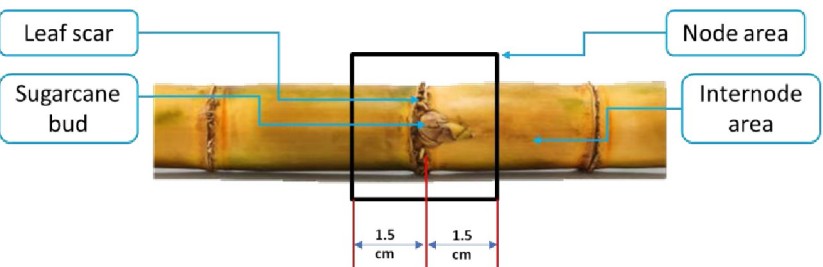

**Fig 1. Composition of sugarcane stalk after cleaning.**

Machine vision technology is a current method used in automatic sugarcane seed cutting machines. There are many problems related to operation, costs, the possibility of commercial application, and the seed quality after cleaning and cutting. Therefore, the current study aims to find an alternative cutting mechanism that overcomes the problems of machine vision technology. The operating process of the proposed machine is as follows: 1. Good and uninfected sugarcane buds are initially identified with a colored pen, thus reducing the cost of the worker required to clean the buds before operation and also avoiding damage to the sugarcane buds during cleaning. In addition, cutting good and uninfected sugarcane buds increases the germination percentage; 2. The sugarcane stalks with pigmented nodes are inserted into the feeding mechanism; 3. The feeding system regulates the speed at which the sugarcane stalks enter the scanning zone, as well as holds the sugarcane stalks during the cutting process; 4. A sugarcane stalk is scanned in the scanning zone using a pair of high-precision RGB color sensors to determine the location of the colored bud to cut the node; 5. The cutting system is driven by a stepper motor, which executes the cutting operation according to the output signals from the control unit.

## 2.2. Overall design of the automatic sugarcane seed cutting machine (ASSCM)

To cut the sugarcane node area with the bud the ASSCM was designed (Fig 2) based on materials available in the local market. In addition, it is characterized by the availability of spare parts and low costs, which reduces the initial and operating costs of the ASSCM. Fig 2 shows six main components of the ASSCM: the frame, the feeding system, the scanning system, the cutting and separation system, and electronic parts.

**2.2.1. Machine frame.** There are a number of vital points that were considered during the design and manufacturing of the machine frame. The final version of the machine frame is lightweight and can be moved depending on the workplace. Fig 3 shows the main dimensions of the machine frame. It also contains wheels through which it is easily moved to meet the worker's comfort. The total height of the machine is about 100 cm, that was adjusted to the worker's vision level while operating the machine for inspection, operation, and maintenance. It contains places for attaching all the electronic parts and units that make up the machine with temporary fastening units (bolts and nuts).

**2.2.2. Feeding system.** The feeding system shown in Fig 3 is designed to accommodate a wide range of sugarcane stalk diameters as well as different degrees of curvature. The feeding system consists of three groups of feeding rollers. Each group consists of a lower roller mounted on ball bearings for easy rotation and an upper roller that is free to move in the vertical axis proportionate to a wide range of sugarcane stalk diameters. A coil spring is used to support the top roller, creating enough pressure to keep the sugarcane stalk in place between the upper and lower rollers. The feeding rollers are constructed using rubber material to prevent any potential harm to the buds while they are being fed and transported. The bottom rollers are driven by a stepper motor, with power being delivered by chains and sprockets. Fig 4 displays the primary components of the feeding system, while Fig 5 provides a detailed depiction of the different viewpoints and proportions of the feeding system.

**2.2.3. Scanning system.** The scanning system is used to determine the locations of the colored buds that need to be cut. As shown in Fig 6, the scanning process takes place inside a closed zone isolated from the surrounding environment to control the intensity of lighting and isolate external influences that could affect the efficiency of the scanning process. The color sensor represents the backbone of the scanning system. Below is a detailed explanation of the color sensor and its operating principle.

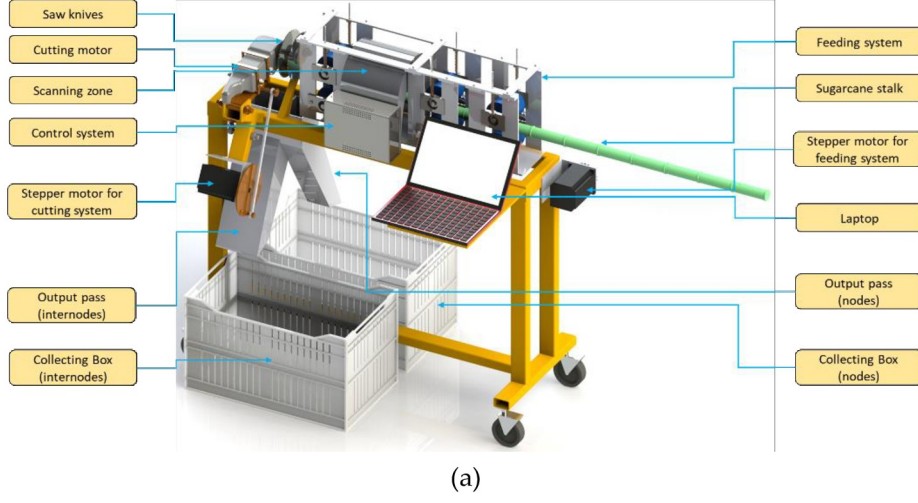

(a)

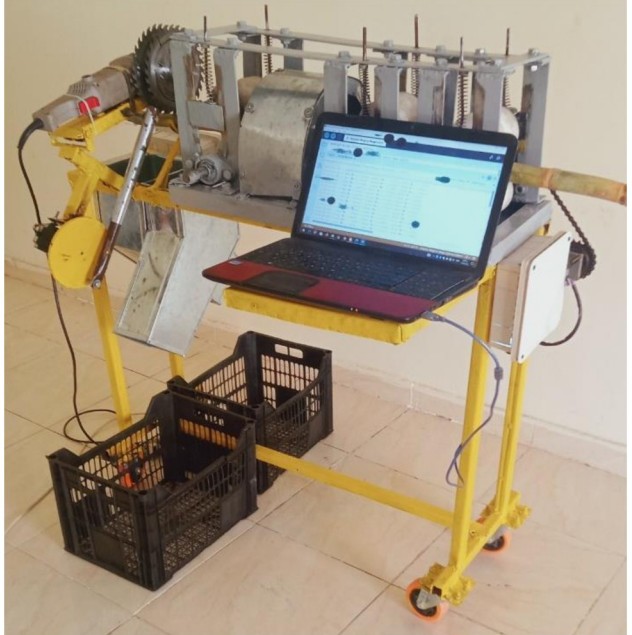

(b)

**Fig 2.** a. Main components of the ASSCM [30]. b. Laboratory test of the ASSCM prototype [30].

Photoelectric color sensors emit light and identify light reflected by an object [31]. For instance, when scanning a colored red substance, every LED shines on the object, causing the reflected light hitting the color sensor. It is commonly understood that a red object typically mirrors all red light that falls on it while absorbing green and blue light, as seen in Fig 7. In this case, the green and blue LEDs reflect minimum light intensity to the color sensor, whereas the red LED reflects maximum light intensity. As a result, the color sensor will have a higher resistance for green and blue light scanning and a lower resistance for red light scanning [32].

**2.2.4. Cutting and separation system.** The cutting system is used to cut the sugarcane nodes (seeds) based on the signal received from the control circuit. The cutting system consists

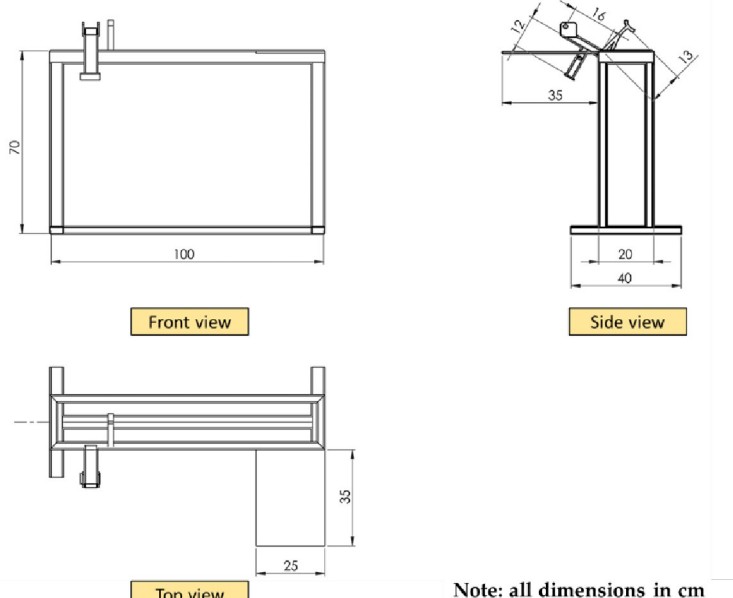

**Fig 3. The main dimensions of the machine frame.**

of several components, as shown in Fig 2. The main component is a 700-watt DC motor (mode: CT13010, crown, China), mounted on a pair of saw blades 7 " (180 mm) in diameter, with 3 cm space between them (the required cutting length, as shown in Fig 1). The vertical movement of the cutting group is controlled by a stepper motor, and the cutting group is provided with a coil spring to facilitate the return of the cutting group after the end of the cutting stroke. For this research, the cutting system is integrated with a pair of output passes to separate the required seeds (nodes) from the rest of the unwanted parts, as shown in Fig 1, each ends with a box for collecting the output parts. Fig 8 shows the operation principle of the sugarcane seed monitoring system integrated through the output path. It is used to count the

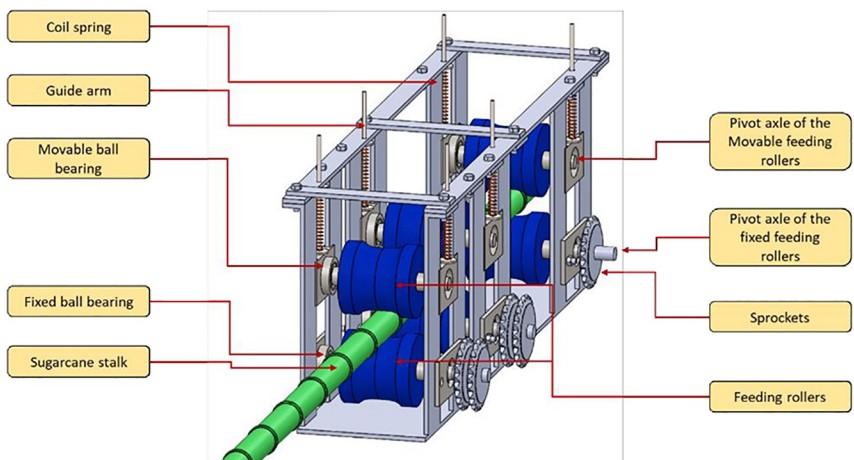

**Fig 4. The main parts of the feeding system.**

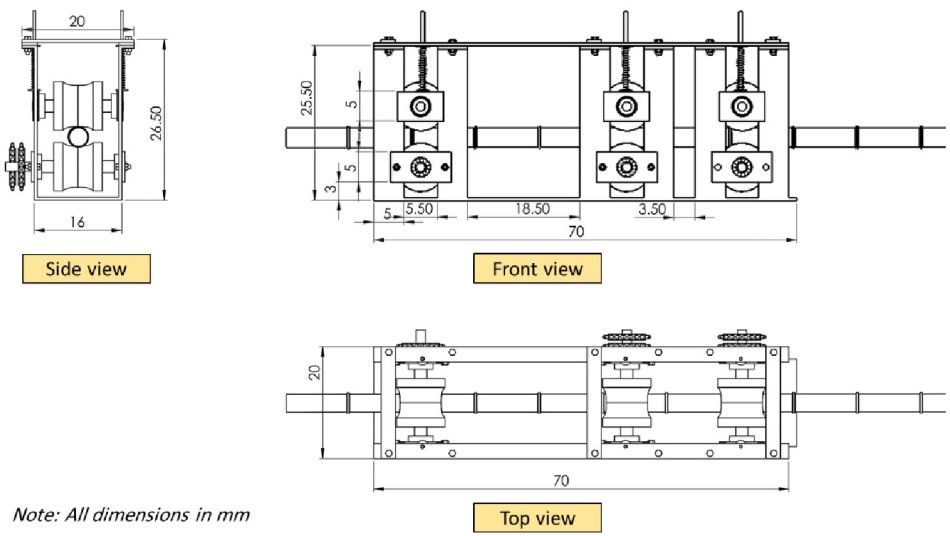

**Fig 5. The detailed views and dimensions of the feeding system.**

sugarcane seeds that have been cut before they fall into the collection box at the bottom of the machine. The sugarcane seed monitoring equipment comprises an ultrasonic sensor positioned along the seed escape pathway. As the sugarcane seeds traverse the pathway, they obstruct the transmission of ultrasonic waves. This obstruction triggers the transmission of a signal to the control unit for data processing. Subsequently, the processed data is sent to the laptop.

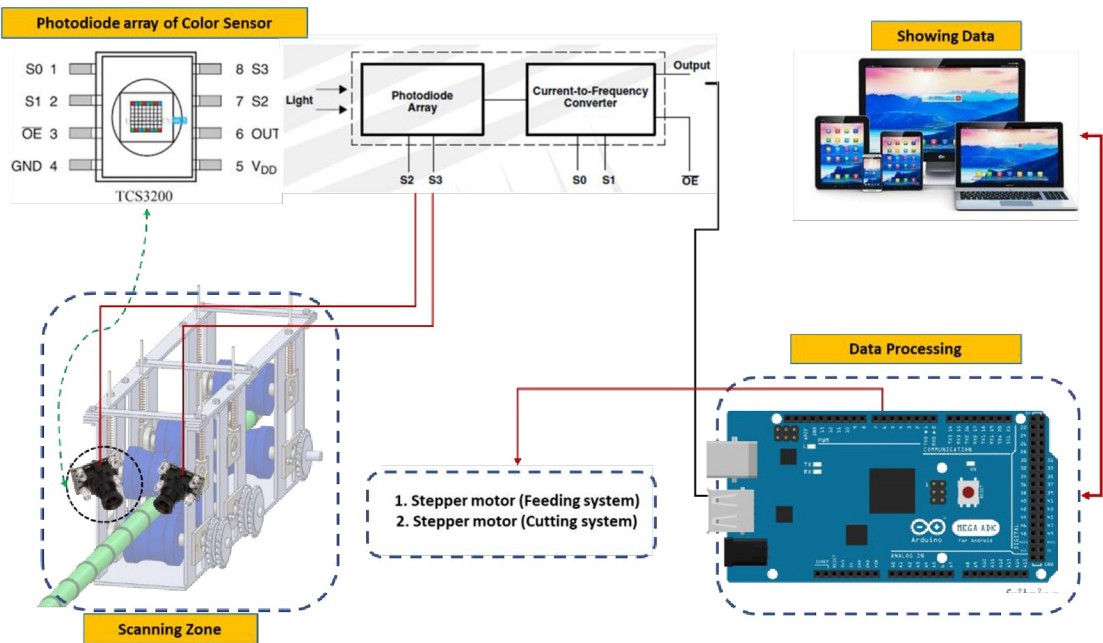

**Fig 6. Signals flow from the color sensor to the data processing unit (Arduino mega board).**

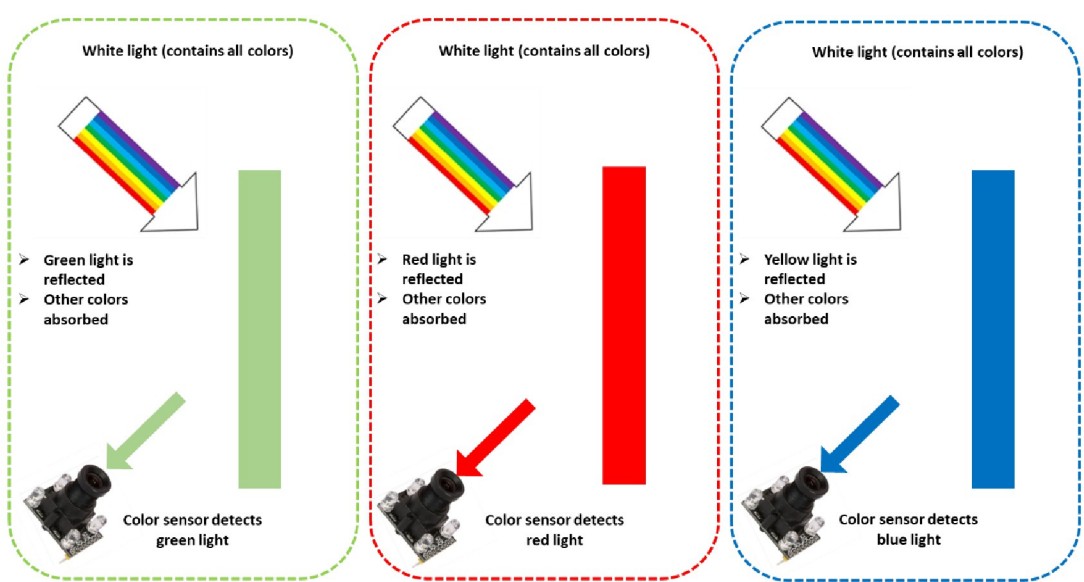

**Fig 7. The principle of the color sensor operation is in measuring the object colors.**

**2.2.5. Electronic parts.** The ASSCM contains many electronic parts, where a stepper motor is used to operate the feeding system and another stepper motor controls the cutting system. The location of the colored buds to be cut is determined using a pair of color sensors, and an ultrasonic sensor is used to count the sugarcane seeds that have been cut before they fall into the collection box. Then, the data is sent to the Arduino Mega board, which controls the operation process based on the operating algorithms. Finally, the data is sent to the user interface using a Wi-Fi module. Fig 9 demonstrates the specifications of different electronic parts used in the current study.

**2.2.6. Electronic circuit design.** The intelligent control and monitoring circuit consists of many parts, such as two-color sensors, an Arduino Mega board (model: TCS3200), two

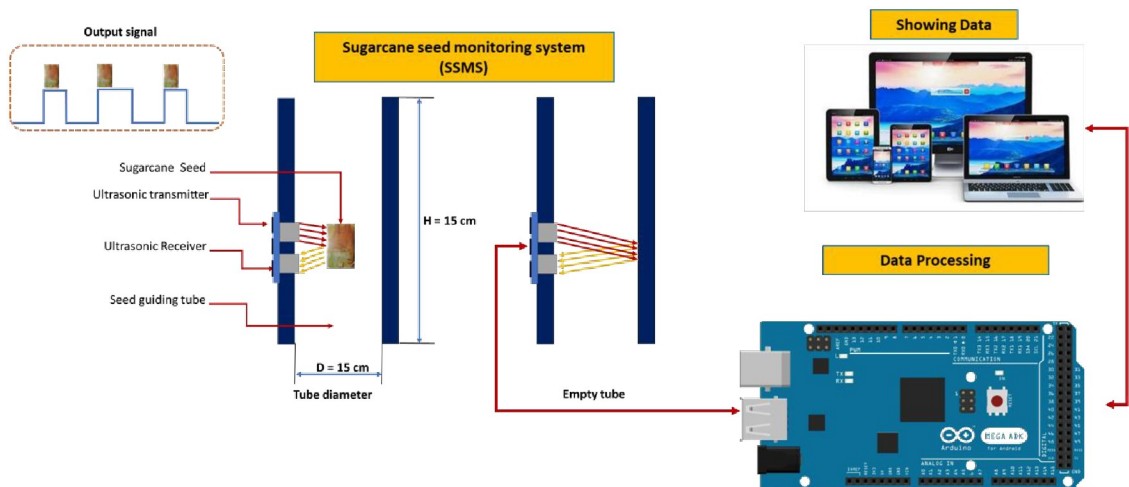

**Fig 8. Operation principle of the sugarcane seed monitoring system.**

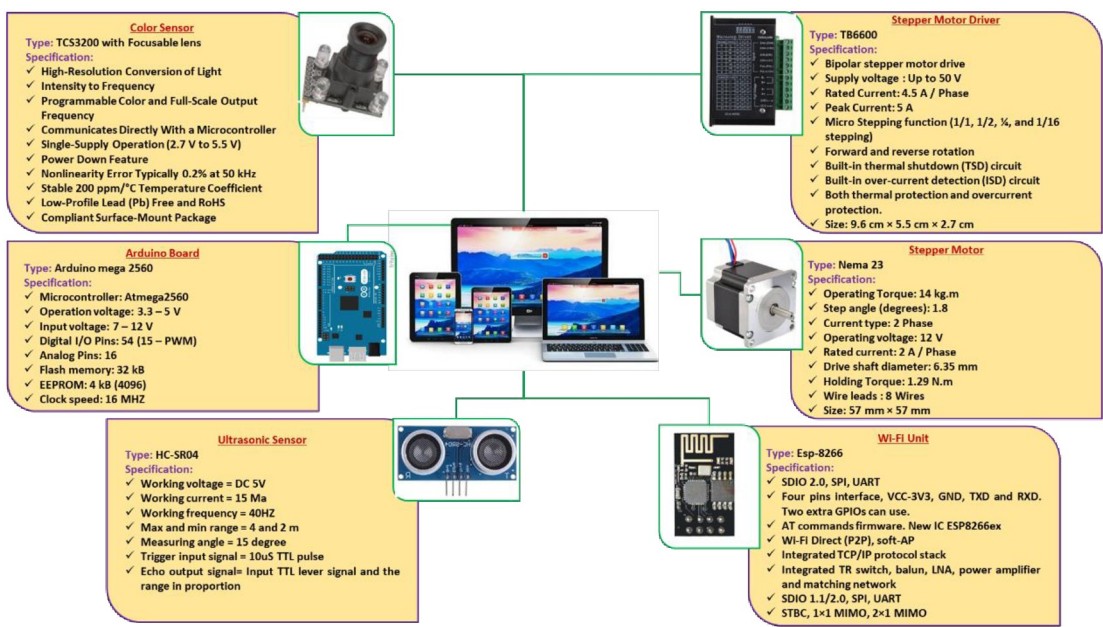

**Fig 9. Specifications of different electronic parts used in the current study.**

stepper motors (model: Nema 23), two motor drivers (model: TB6600), an ultrasonic sensor (model: HC-SR04), Wi-Fi module (model: ESP-8266), and a PV system (1000 watt). Fig 10 shows the correct electrical connections for various electronic components used in the current study.

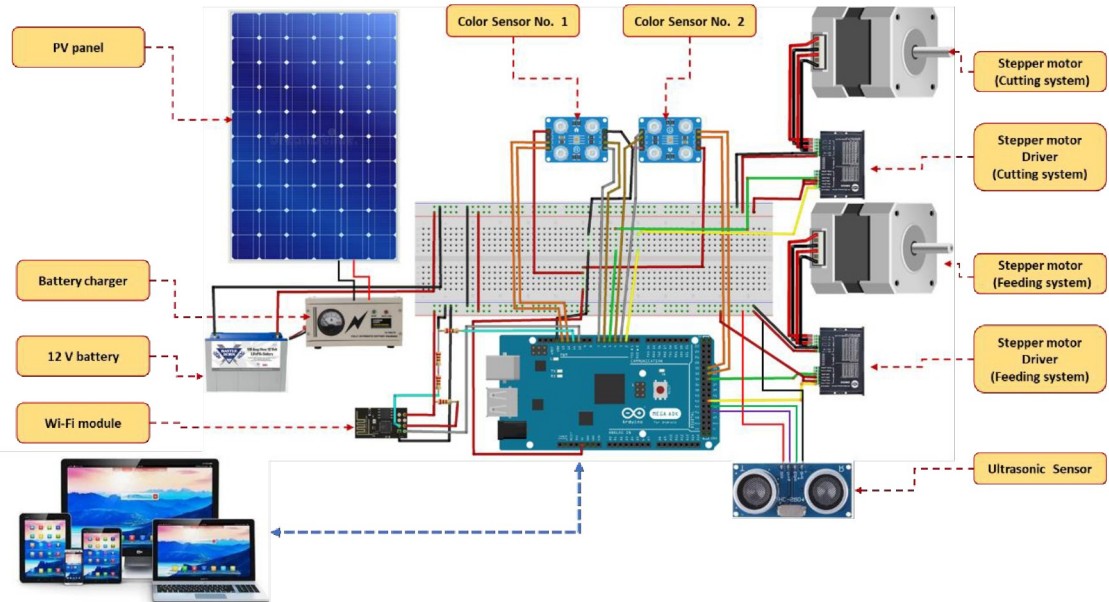

**Fig 10. The correct electrical connections for various electronic components used in the current study.**

## 2.3. Operating algorithms

**2.3.1. Operating algorithm of the control system.** Fig 11 depicts the operational algorithm used in the ASSCM's optical scanning system and autonomous cutting system. Prior to uploading the programming code to the Arduino Mega board, the code is adjusted according to specific operational criteria, such as the desired speed of the feeding system, the delay between the scanning and cutting processes, the positioning of the color sensor lenses, and the speed of the cutting system. Subsequently, the programming code is transferred to the Arduino Mega board, and at the start of operation, the stepper motors and color sensors are initialized. Subsequently, the feeding mechanism is activated, allowing the sugarcane stalk with pigmented nodes to traverse the scanning area. The resulting signal from the color sensors is then sent to the Arduino Mega board for analysis and subsequent decision-making, guided by predetermined values. Using the output signals as criteria, there are two possible actions: if the output signals match the predetermined value, the Arduino board will command the feeding system to stop and activate the cutting system. Otherwise, the feeding system will continue operating while the cutting system remains in a standby state.

**2.3.2. Operating algorithm of the sugarcane seed monitoring system.** Fig 12 depicts the operational algorithm of the sugarcane seed monitoring system. Under normal circumstances, when the output passage is vacant, the ultrasonic sensor measures a distance of around 15 cm. If a sugarcane seed traverses the output path, it obstructs the ultrasonic waves, resulting in a recorded distance of less than 15 cm. The work process involves the generation of an electronic impulse when the sugarcane seed is detected by the sugarcane seed counter sensor. A software program is then executed to count the number of sugarcane seeds. The final count is transferred to the communication serial port following a specific communication protocol. The communication serial port is connected to the Wi-Fi module (ESP-8266), which enables the collected data to be sent to the laptop software for real-time display.

**2.3.3. Operating algorithm of the Wi-Fi module.** Fig 13 shows the operating algorithm for transporting analog data via a Wi-Fi module that collects data from different sensors and

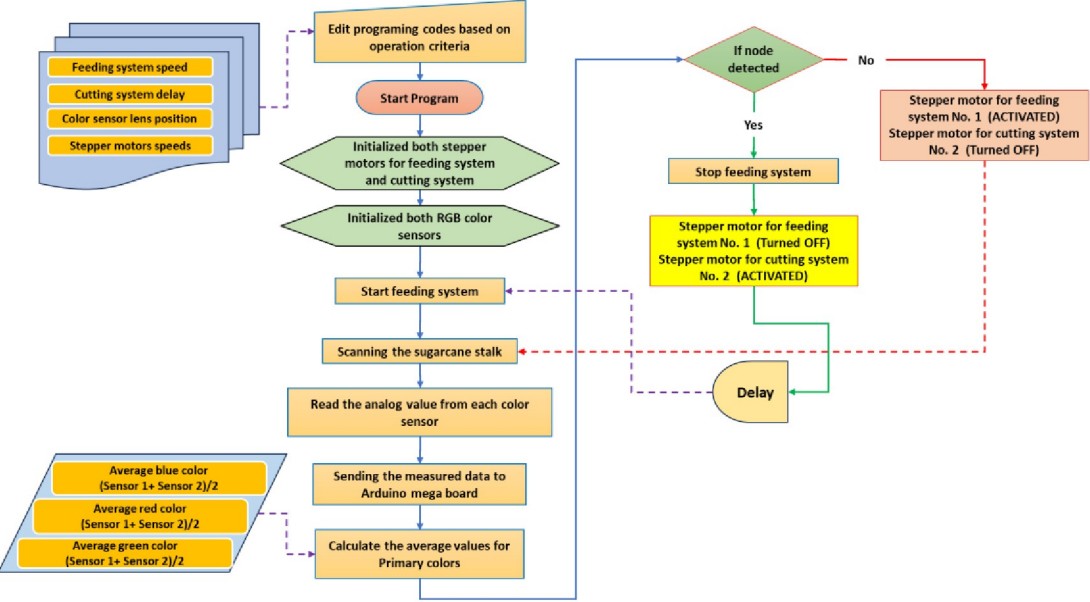

**Fig 11. The operating algorithm for the optical scanning system and automatic cutting system used in the ASSCM.**

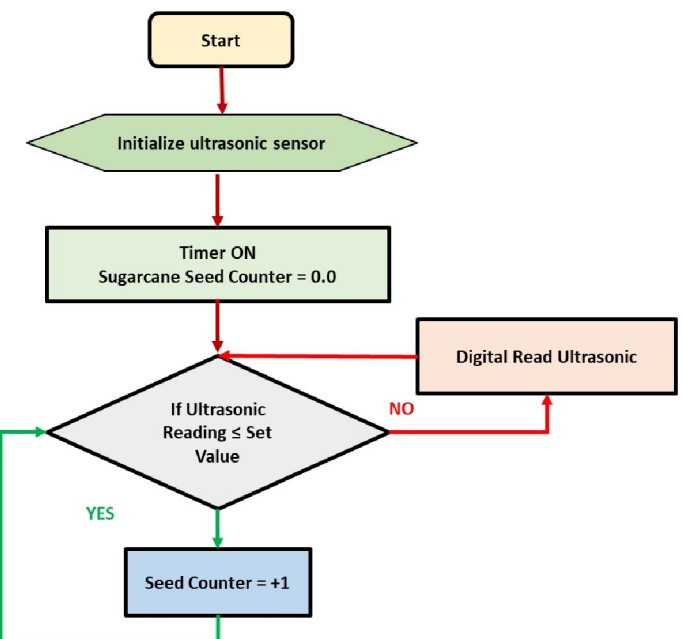

**Fig 12. Operating algorithm of the sugarcane seed monitoring system.**

then sends it to the user interface (smart phone or laptop) via the internet using a Wi-Fi unit. It starts by initializing the Arduino Mega board and the Wi-Fi module. Then, it tries to establish a connection to the internet. If the connection is established successfully, the program generates an IP address and transports the analog data.

## 2.4. Sensor's calibration

The calibration tests of the RGB color sensors were conducted under laboratory conditions in the Department of Agricultural Engineering, Aswan University. To reduce damage of the sugarcane buds during the cutting process and obtain the best performance of the machine, calibration process must be performed for the color sensors used to detect buds and the ultrasonic sensor used to count the sugarcane seeds. The calibration process was conducted as follows:

**2.4.1. Color sensor calibration.** The both color sensors (mode: TCS3200) were mounted on the scanning zones in opposing directions for calibration and sensing the RGB color channels at a prescribed height of 30 mm above the sugarcane stalk, under 25 lux light intensity, as described by [30, 33, 34]. During the calibration of color sensors, three different color pens were used for coloring the sugarcane nodes [Red, blue, and black]. Both lowest and highest RGB values for each color channel were recorded. The obtained results from the calibration of the color sensors were used for editing the programming codes of the scanning system. All calibration tests were replicated at least three times.

**2.4.2. Calibration of the ultrasonic sensor.** This test is about calibrating the ultrasonic sensor of the sugarcane seed monitoring system to verify the counting accuracy of different spacing and estimate the relative error. The calibration process was done by placing an object at variable distances from the ultrasonic sensor and comparing the distance measured by the ultrasonic sensor to the distance measured by the traditional method, as stated by Elwakeel et al. [35]. All calibration tests were replicated at least three times.

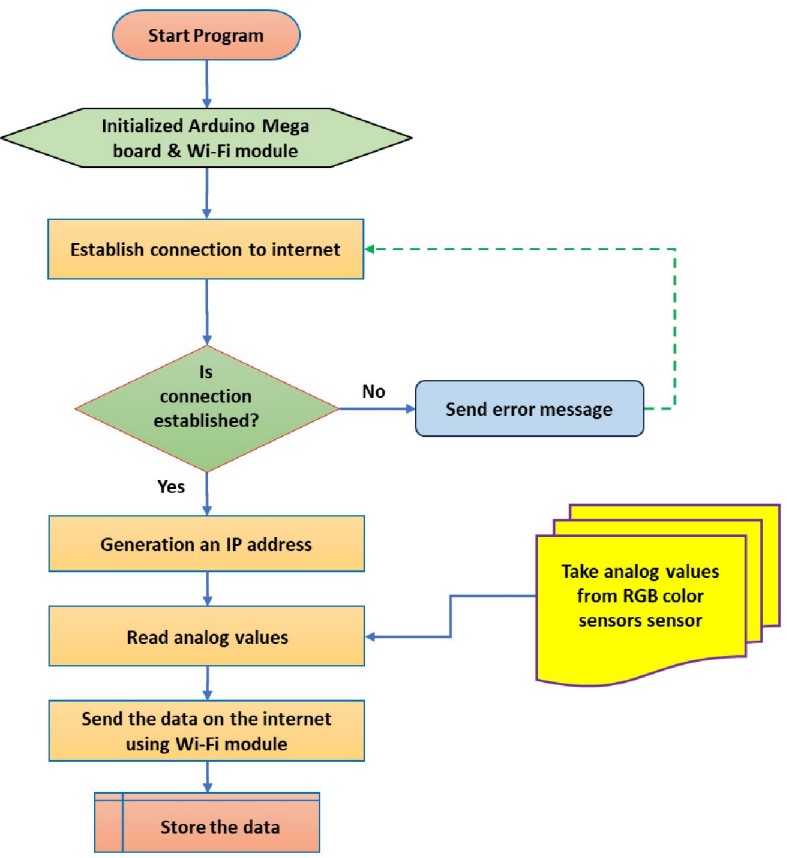

**Fig 13. Operating algorithm of the Wi-Fi module.**

## 2.5. Estimation of damage index

The performance tests of the ASSCM were conducted under laboratory conditions in the Department of Agricultural Engineering, Aswan University. The quality of the sugarcane seeds was determined as a function of sugarcane stalk diameter, cutting time, and the type of the saw knives. All tests were done at a fixed cutting speed of 12000 rpm. Where increasing the speed of cutting tools may affect their performance. But in a study by Martins and Ruiz [36] on sugarcane crops, no significant change in raw material losses was observed with respect to operational speed. However, Filho et al. [37] found that increasing harvest speed can lead to increased losses from raw materials. In this particular trial, the harvester speeds did not vary significantly. Also, four different diameters of the sugarcane stalk ($d_1$ = 2.03, $d_2$ = 2.72, $d_3$ = 3.42, and $d_4$ = 3.94 cm), where the cutting force, cutting power and damage index depending directly on the sugarcane diameter [30, 38–40]; two different types of the saw knives with the same diameter (7.0 in/180 mm) ($SN_1$ = 30 teeth, and $SN_1$ = 80 teeth), where the cutting tool is one of the main causes of damage during harvesting [38, 41, 42]. Also, Samaila et al. [38] and Paulo Testa et al. [42] reported that using of rotary saw knives obtained better cut quality compared to impact knives. In addition, Mello and Harris [43] stated that saw knife with small pitch make penetration easer and cutting more effective; and five cutting times ($T_1$ = 1000, $T_2$ = 1500, $T_3$ = 2000, $T_4$ = 2500, and $T_5$ = 3000 ms), the cutting time have higher effect of the quality of the cutting process as stated by [42, 44]. Each test was repeated at least three times.

| Classification | Lower edge | Upper edge | Weight |
|---|---|---|---|
| Without damage (SD) | | | -1,00 |
| Partial damage (PD) | | | 0,00 |
| Extreme damage (ED) | | | 1,00 |

**Fig 14. Classification of damage caused to sugarcane seeds, according to [43, 46].**

At the end of each test, the collected sugarcane seeds were classified according to [45, 46], as shown in Fig 14. Researchers [41, 46, 47] stated that damage index values were classified into three sections: without damage (damage index $< -0.33$), partial damage (damage index from —0.33 to 0.33), and extreme damage (damage index $> 0.33$).

Each classification represents a weight used for the calculation of the ID according to [37, 46], and the ID was calculated using Eqs 1 and 2.

$$ID = \frac{P_{SD}.n_{SD} + P_{PD}.n_{PD} + P_{ED}.n_{ED}}{n} \tag{1}$$

$$n = n_{SD} + n_{PD} + n_{ED} \tag{2}$$

where: $P_{SD}$, $P_{PD}$, and $P_{ED}$ are the weight attributed (without damage, partial damage, and extreme damage, respectively); $n_{SD}$, $n_{PD}$, and $n_{ED}$ are the number of sugarcane seeds (without damage, partial damage, and extreme damage, respectively), and $n$ is the total number of sugarcane seeds.

## 2.6. Economic analysis

The economic analysis of the ASSCM was performed to determine the commercial sustainability and viability. The Egyptian financial environment served as the basis for estimating the economic performance metrics. The economic performance parameters annualized cost of payback period and net present value as the key performance indicator parameters based on the findings of [48–50].

The annualized investment cost ($C_a$) of the ASSCM was calculated using parameters in Eq 3,

$$C_a = C_{ac} + C_m - V_a \tag{3}$$

where, $C_{ac}$ is the annualized capital cost, $C_m$ is the maintenance costs, 3% of the annual capital cost, and $V_a$ is the salvage value, 8% of the annual capital cost.

$$C_{ac} = C_{cc} \times F_c \tag{4}$$

$$F_c = \frac{d(1+d)^n}{(1+d)^n - 1} \tag{5}$$

where, $C_{cc}$ is the total capital cost, $F_c$ is the capital recovery factor, $d$ is interest rate (equal 20%), $n$ is the operating life, 5 years for the ASSCM and 20 years for the PV system.

The sugarcane seed cutting cost per hectare ($C_s$) is calculated as in [48–50].

$$C_s = \frac{C_a}{M_y} \qquad (6)$$

The number of hectares planted per year ($M_y$) is calculated as

$$M_y = M_d \times D / n \qquad (7)$$

where, $M_d$ is the amount of sugarcane seed cutting per day, and $D$ is the number of days the ASSCM operates in a year (Assuming D = 90 day), n is the number of sugarcane seed required for planting one hectare (Assuming the planting spacing is 1.0 m pacing between rows and 0.5 m spacing between plants in the same raw, so n = 20000 sugarcane seed).

The total cost of sugarcane seeds per hectare is calculated as stated by [48–50].

$$C_{ds} = C_{dp} + C_s \qquad (8)$$

where, $C_{dp}$ is the cost of fresh sugarcane stalks per hectare, which is calculated as

$$C_{dp} = C_{fd} \times M_f \qquad (9)$$

where, $M_f$ is the quantity of fresh sugarcane stalks per hectare, kg and $C_{fd}$ is the cost of fresh sugarcane stalks per kg.

Savings obtained per hectare of sugarcane seeds ($S_{ha}$) is given by Eq 10.

$$S_{ha} = SP_c - C_{ds} \qquad (10)$$

where $SP_c$ is the selling price of sugarcane seeds per hectare.

The saving obtained from the ASSCM per day ($S_d$) is given by Eq 11.

$$S_d = S_{ha} \times M_y \qquad (11)$$

The saving obtained from the ASSCM per year ($S_y$) is given by Eq 12.

$$S_y = \frac{S_{ha}}{D} \qquad (12)$$

The payback time (N) for the ASSCM is calculated as recommended by [48–51].

$$N = \frac{ln\left[1 - \frac{C_{cc}}{S_1}(d - i)\right]}{\ln\left(\frac{1+i}{1+d}\right)} \qquad (13)$$

where, $i$ is the inflation rate (equal 39.7%), and $S_1$ is the saving obtained from ASSCM after the first year.

## 2.7. Statistical analysis

The statistical metrics were employed to confirm the reliability of the used sensors via comparing the values obtained by the sensors to the reference value using standard instruments. The determination coefficient (R) and root mean square error ($R^2$) were among the measures used. The statistical analyses (ANOVA) were carried out using the IBM SPSS version 25 statistical analysis program and Microsoft Office, in particular Excel 365.

# 3. Results and discussions

## 3.1. Sensor's calibration

### 3.1.1. Calibration of the color sensors.
Three different sheets colored by red, blue, and black were used for calibrating the color sensors. These three colors represent the candidate colors that will be used to color the sugarcane buds to be cut. To achieve excellent detection efficiency of sugarcane buds, the RGB color channels of the color sensors were calibrated. Furthermore, the Color-based RGB sensors were additionally calibrated to produce the highest and lowest RGB values for each color. These values were then used to modify the programming code for the color sensors (scanning system). Fig 15 shows the average values of the RGB channels obtained from color sensor laboratory testing, where the speed of the feeding system was zero (stopped) at 30 lux light intensity. The obtained results come in agreement with [30, 32–34, 52]. According to Khanh et al. [53], due to space limitations, the sensor cannot be placed closer than 15 mm. Elwakeel et al. [34] reported that the radiation intensity was stable when the sensor is placed at a distance higher than 30 mm. also, they stated that the difference between the green and blue channels can be clearly seen at a light intensity level from 25–40 lux. This is explained by the fact that increased light intensity increases the reflectance and reduces RGB color pulse width (PW) [54, 55].

### 3.1.2. Calibration of the ultrasonic sensor.
The acoustic standing wave sensor is used in the sugarcane seed monitoring system. To obtain good results, the sensor must be calibrated to know the value of $R^2$ to modify the programming code. The measured distance using the ultrasonic sensor (model: HC-SR04) was calibrated with the observed measurements by the length scale. The obtained calibration results are presented in Fig 16. The results revealed an excellent relationship between the recorded and reference distance values, having a high $R^2$ of 0.96, showing a good match between the measured and calculated distance and the reference distance. The ultrasonic sensor (model: HC-SR04) performed perfectly at all distance values tested and exhibited good linear regressions that comes close to fitting with the 1:1 line (y = x + 0). To forecast the measured distance d, the following linear regression equation was developed. Elwakeel et al. [35] stated that the using of Ultrasonic sensor in seedlings couniting give high and accurate performance of the transplanter machine. In addition many researchers such as [56–59], used the Ultrasonic sensor (model: HC-SR04) for object detection, where they reported that the detection process was very high and acceptable.

$$y = 1.0027\,x - 0.1784 \qquad (14)$$

## 3.2. Classification of sugarcane stalks used in the current study

For the laboratory tests, a total of 100 sugarcane stalks with different diameters were harvested from a local farm located in Aswan, Egypt. Performance tests were conducted during the period of December 1, 2023, to December 12, 2023. After that, sugarcane stalks were divided into four categories according to stalk diameters, as shown in Table 1. Where the present study utilized sugarcane sticks that were segregated into four groups to assess the performance of the machine. The efficiency of each group was evaluated using the analysis of variance (ANOVA) test. The results of the test indicated that there were significant differences among the four groups, thereby affirming the efficacy of the grouping method in assessing the performance of the machine.

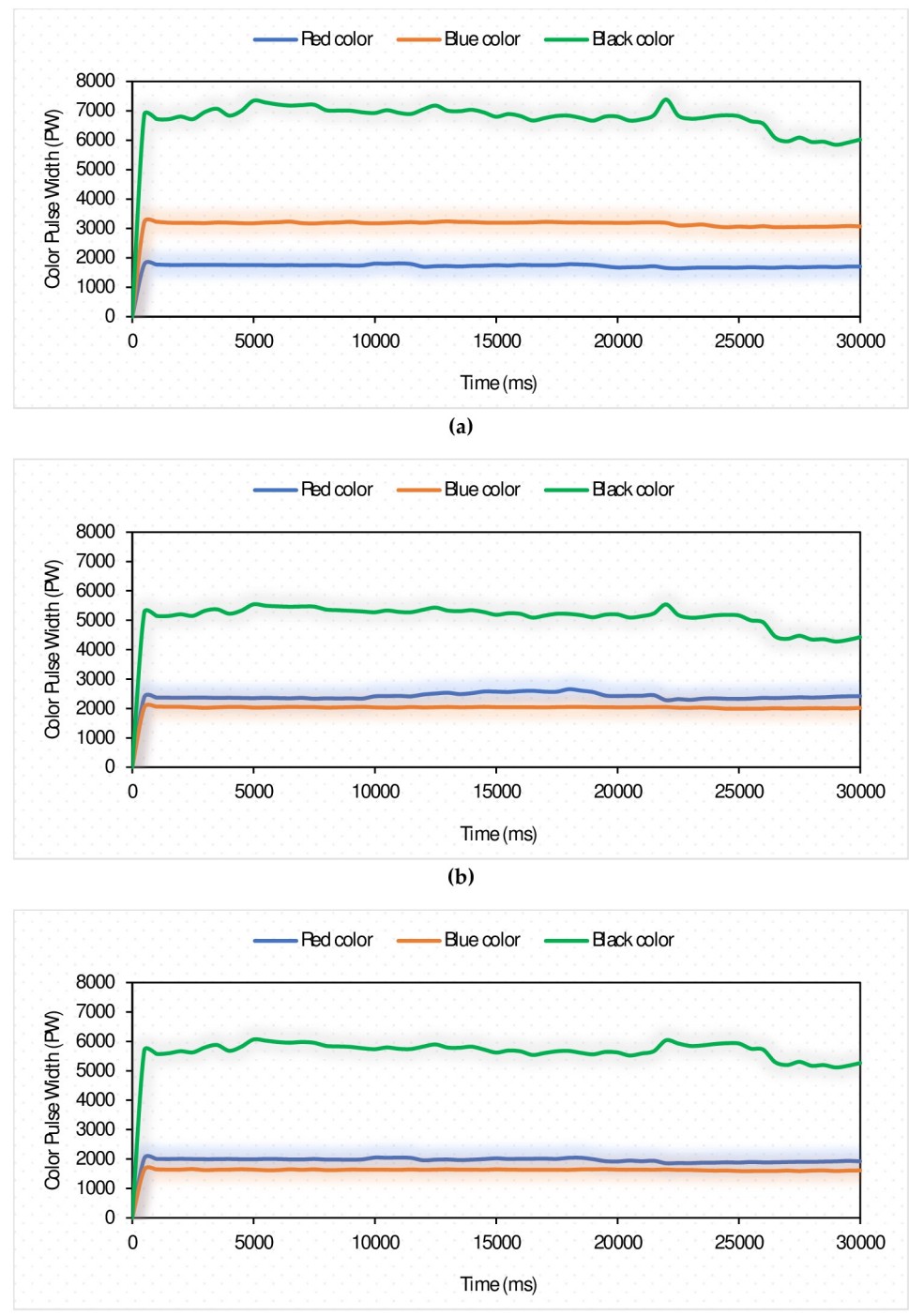

**Fig 15.** Calibration of the color sensors: a. red channel; b. green channel; and c. blue channel.

## 3.3. Damage index (DI)

For years, several researchers around the world have studied the relationship between the cutting quality and the cutting mechanism's design [42, 46, 60–66]. The cutting mechanism's performance is affected by a number of cutter design parameters, including cutting velocity, the

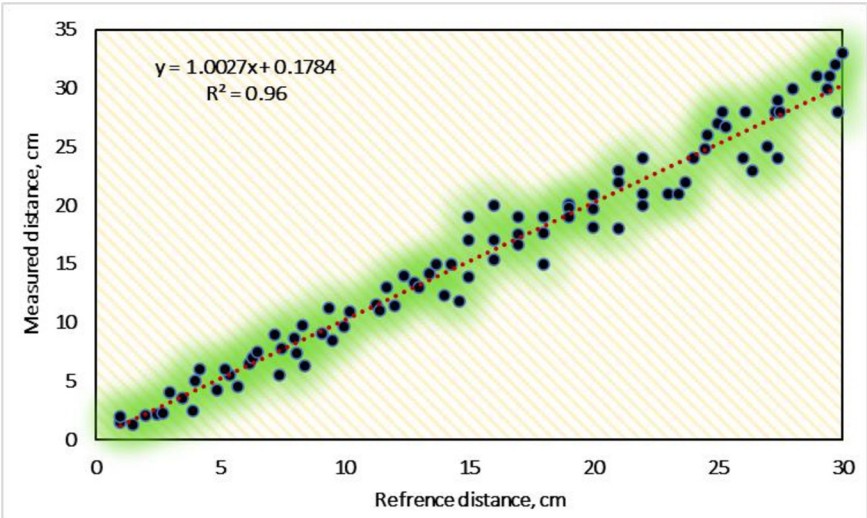

**Fig 16. Calibration results for ultrasonic sensor.**

overlapping length of the upper and lower chopping blades (harvesting choppers), the blade's thickness, coating, and the overlapping length of the upper and lower chopping blades (harvesting choppers), among others [43, 62, 67]. As mentioned, the quality of the sugarcane seeds as a function of sugarcane stalk diameter, cutting time, and the type of the saw knives were estimated considering four different diameters of the sugarcane stalk ($d_1$ = 2.03, $d_2$ = 2.72, $d_3$ = 3.42, and $d_4$ = 3.94 cm), two different types of the saw knives have the same diameter (7.0 in / 180 mm) ($SN_1$ = 30 teeth, and $SN_1$ = 80 teeth), and five cutting times ($T_1$ = 1000, $T_2$ = 1500, $T_3$ = 2000, $T_4$ = 2500, and $T_5$ = 3000 ms). All tests were done at a fixed cutting speed of 12000 rpm.

The data obtained are shown in Fig 17. The saw knives with 80 teeth had DI values below 0.00 (indicating little damage) for all cutting periods and sugarcane stalk diameters tested. This suggests that the cutting knives have a high cutting quality compared to the cutting systems. However, while examining saw blades with 30 teeth, it was found that the majority of DI values were below 0.00 for all cutting periods and sugarcane stalk diameters during testing. The Damage Index (DI) values were zero, indicating partial damage, for sugarcane stalks with a diameter of 3.42 cm during cutting periods of 2000 ms and 2500 ms. In addition, the Damage Index (DI) values were found to be zero (indicating extreme damage) for sugarcane stalks with a diameter of 3.94 cm when cut at 1500 ms and 2000 ms. These results are comparable to those reported by [45, 64,], who found a DI value of -0.6 for straight blades, thus confirming the findings of this study. The DI values stayed below 0.00, indicating partial damage and adequate performance of the cutting devices. However, this may be attributed to the fact that they are

**Table 1. Classification of sugarcane stalks used in the current study based on stalk diameters.**

| Group no. | Average stalk diameter, cm | Standard deviation | Mean stander Error | Variance |
|---|---|---|---|---|
| G1 | 2.03 | 0.246 | 0.087 | 0.061 |
| G2 | 2.72 | 0.233 | 0.078 | 0.054 |
| G3 | 3.42 | 0.216 | 0.063 | 0.047 |
| G4 | 3.98 | 0.183 | 0.050 | 0.034 |

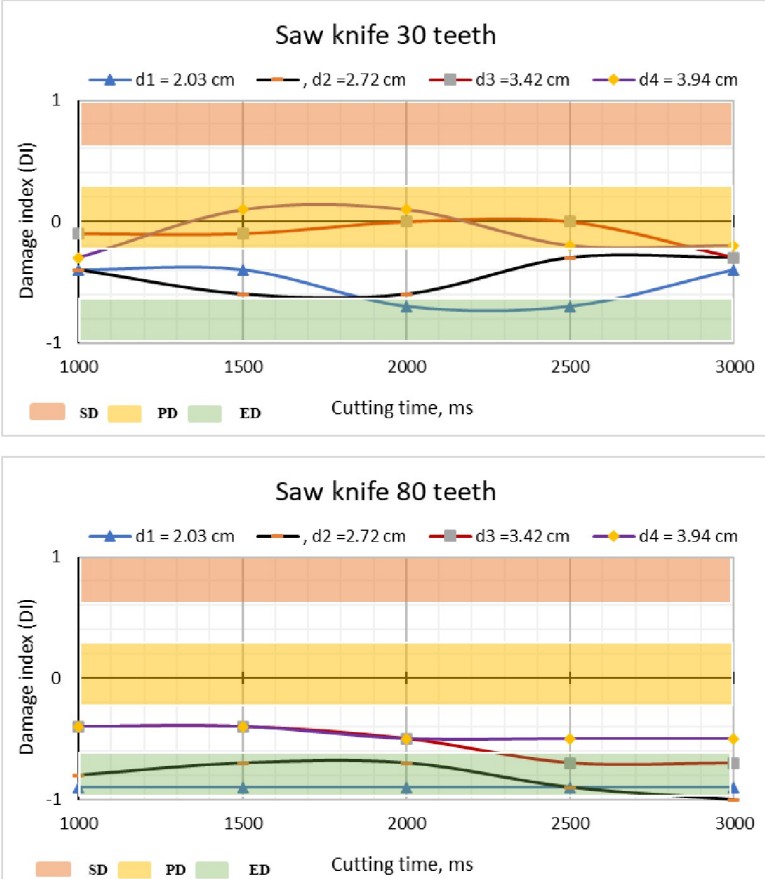

**Fig 17. The relationship between ID and the cutting time, where SD means without damage, PD means partial damage, and ED means extreme damage.**

recently introduced cutting instruments with little use, as mentioned by [42]. Fig 17 illustrates the correlation between ID and the duration of the cutting process. The highest DI values were found to occur when the cutting duration was shorter than 2000 ms and the mean diameter was more than 2.72 cm. The DI values fell as the cutting time increased, eventually reaching a minimum value of 3000 ms. The maximum and minimum values of DI were observed at average diameters of 3.94 cm and 2.03 cm, respectively. Furthermore, using a cutting duration of 3000 ms yielded the least amount of damage in comparison to the other cutting periods, as seen in Fig 17.

Momin et al. [63] studied the effects of four base cutter blade designs on sugarcane stem cut quality. Where they reported that the percentages of undamaged stems for were ranged between 62.1%, and 83.1%; partially damaged stems were ranged between 11.25%, and 17.73%; and extreme damaged stems were ranged between 11.9%, and 15.9%. Elwakeel et al. [68] developed and assessed a partially automated machine for cutting sugarcane buds. The machine was tested at various cutting speeds, with different types of cutting blades, and on stalks of varying diameters. The results indicated that the damage index and invisible losses fell within acceptable limits (-1.0 to 0.0) for all tested variables. Additionally, it was reported that the highest and lowest values of the damage index were observed at mean diameters of 2.8 cm and 1.32 cm, respectively. Also, Paulo Testa et al. [42] stated that using of rotary saw knives

can led to decrease the DI by about 15–28% compared with impact knives. And reducing the visible losses by about 74%. Filho et al. [37] demonstrated that using of rotary saw give acceptable levels for the sugarcane stalk DI and decreased general damage.

## 3.4. Damage frequency

Fig 18 displays the damage frequency of the sugarcane seeds, given as a percentage. Considered were five cutting times ($d_1$ = 2.03, $d_2$ = 2.72, $d_3$ = 3.42, and $d_4$ = 3.94 cm), two different types of saw knives with the same diameter (7.0 in/180 mm) ($SN_1$ = 30 teeth, and $SN_1$ = 80 teeth), and five cutting times ($T_1$ = 1000, $T_2$ = 1500, $T_3$ = 2000, $T_4$ = 2500, and $T_5$ = 3000 ms), all tests were done at a fixed cutting speed of 12000 rpm. The obtained results showed that no partial or extreme damage was recorded in all cutting buds, while partial damage was recorded only for sugarcane stalk diameter of 3.42 cm at cutting times of 2000 ms and 2500 ms. Furthermore, the severe destruction was observed exclusively in sugarcane stalks with a diameter of 3.94 cm when cut at 1500 ms and 2000 ms. Conversely, sugarcane seeds remained completely undamaged when the stalk diameter was 2.72 cm and the cutting time was 3000 ms, demonstrating

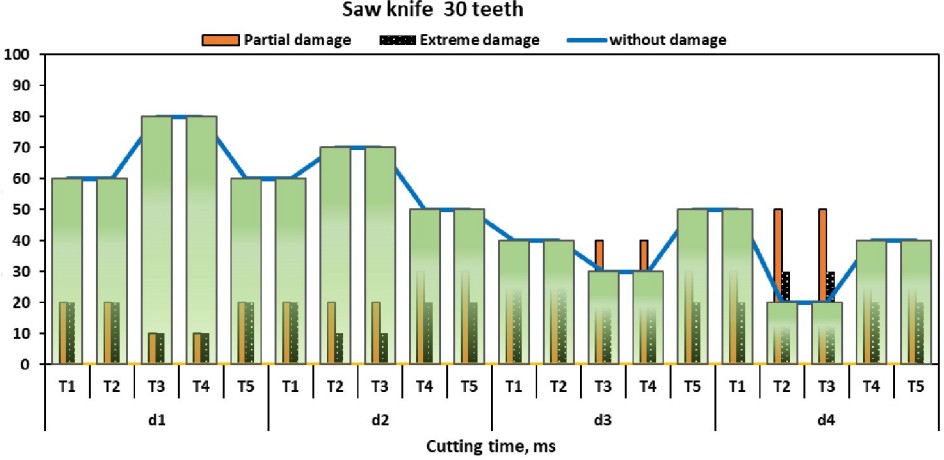

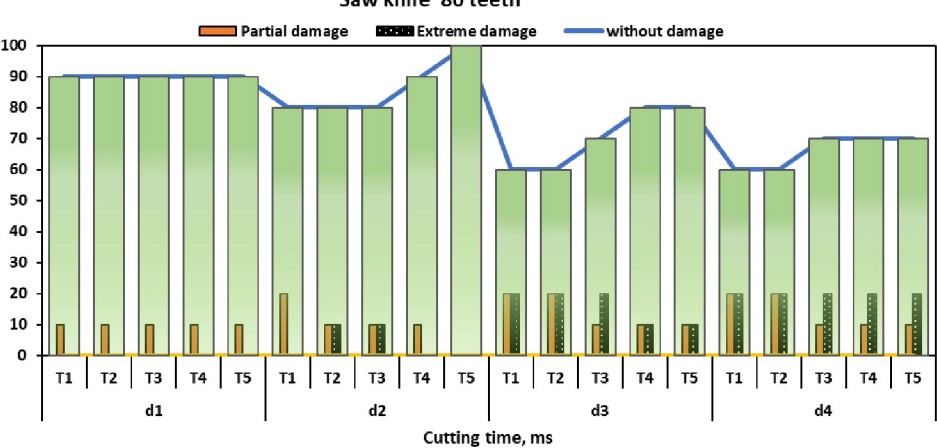

**Fig 18. Frequency of damage as a function of cutting time, where $d_1$ = 2.03 cm, $d_2$ = 2.72 cm, $d_3$ = 3.42 cm, $d_4$ = 3.94 cm, $T_1$ = 1000 ms, $T_2$ = 1500 ms, $T_3$ = 2000 ms, $T_4$ = 2500 ms, and $T_5$ = 3000 ms.**

the cutting system's optimal efficiency. The proportion of intact buds rose, but the proportion of buds with moderate damage and severe damage decreased with each subsequent cutting.

Fig 18 demonstrates that using saw blades with 30 teeth results in an escalation of both partial damage and severe damage in the separated sugarcane seeds. Similarly, increasing the stalk width has the same effect. The test results consistently showed the least amount of damage while using a saw knife with 80 teeth, a cutting duration of 3000 ms, and a stalk diameter of 2.03 cm. According [69] the sugarcane stalk may have experienced external force while being cut, leading to splits and lacerations that increased the occurrence of severe damage. During the process of cutting a vegetable, the fibers are compressed and pushed in the direction of the cutting tool, causing sequential ruptures as the tool moves ahead. This observation supports the findings of variability in the results of this research. Furthermore, [70] demonstrated that there was a positive correlation between the diameter of the sugarcane stalk and the frequency of damage. The lowest frequency of damage was seen when the stalk diameter reached 1.32 cm.

## 3.5. Economic analysis

The ASSCM's annual cost is determined by its capital cost, maintenance cost, operational cost, and salvage value [50]. Table 2 shows all the costs related to the ASSCM. The ASSCM has a high capital cost due to the use of electronic components and a PV system. Due to this, the annual cost of ASSCM.

The economic parameters that depend on the ASSCM and PV system are shown in Table 3. The payback time of the ASSCM depends on the capital cost of the ASSCM and PV system and the savings obtained from the ASSCM per year [50]. The obtained data in the same table also shows that the number of hectares can be planted per day using the ASSCM is 0.45 and

**Table 2. Different costs related to the ASSCM and PV system.**

| Cost parameters | ASSCM | PV system |
|---|---|---|
| Capital cost, USD | 285.7 | 571.43 |
| Annual capital cost, USD | 21.50 | 1.31 |
| Annual maintenance cost, USD | 8.571 | 17.14 |
| Annual salvage value, USD | 22.86 | 45.71 |
| Annual cost, USD | 52.93 | 64.16 |

**Table 3. Economic analysis of the ASSCM and PV system.**

| Economic analysis parameters | |
|---|---|
| No. of sugarcane seeds cutting per day | 9000 |
| No. of sugarcane seeds required per hectare | 20000 |
| No. of hectare seeds can be cut per day | 0.45 |
| Cost of cutting sugarcane seeds per hectare, USD | 2.84 |
| Cost of fresh sugarcane stalks per hectare, USD | 68.025 |
| The total cost of sugarcane seeds per hectare, USD | 70.865 |
| Selling price per hectare, USD | 120 |
| Saving per hectare, USD | 49.135 |
| Saving per day, USD | 22.111 |
| Saving after 1 year, USD | 1990 |
| Payback time (Years) | 0.536 |

**Table 4. Comparison of the ASSCM with other technologies *.**

| Ref. | Operating technology for fully automatic machines | Detection time per bud (sec) | Recognition rate, % | Damage index | Damage frequency | Owning cost, USD | Operating cost, USD | Sugarcane stalk property | Node detection |
|---|---|---|---|---|---|---|---|---|---|
| [5] | Wavelet analysis | 0.25 | 99.63 | N.D. | N.D. | N.D. | N.D. | CWL | G&I |
| [13] | Image processing & machine vision | 0.5 | 80 | N.D. | N.D. | N.D. | N.D. | CWL | G&I |
| [16] | Machine vision | 0.539 | 93 | N.D. | N.D. | N.D. | N.D. | CWL | G&I |
| [27] | Yolov$_3$ network | 0.028 | 90.38 | N.D. | N.D. | N.D. | N.D. | CWL | G&I |
| [71] | Wavelet analysis | 0.21 | 98.5 | N.D. | N.D. | N.D. | N.D. | CWL | G&I |
| [19] | Maximum value points of the vertical projection function | - - | 95 | N.D. | N.D. | N.D. | N.D. | CWL | G&I |
| Proposed ASSCM | IoT and RGB color sensors | 1.0–1.75 | 95–100 | < 0.00 | 100% without damage | 857.13 USD | 70.865 USD per hectare | With leaves | Good buds only |

* CWL means clean without leaves; G&I means good and infected; and N.D. means not detected.

the total cost of sugarcane seeds per hectare is 70.865 USD. In addition, the ASSCM can pay for itself in a short period of time. The payback time is 0.536 years, which means that the ASSCM will save enough money to pay for itself in about 6.43 months. Overall, the table shows that the ASSCM is an economical way to plant sugarcane. In addition, due to the lack of economic studies for the automatic sugarcane seed cutting systems, it is impossible to determine the economic efficiency of the manufactured machine in comparison to other systems worldwide. So, this study aimed to provide an economic analysis of the machine, which can be used as a reference point for evaluating and comparing the automatic sugarcane seed cutting systems in the future. On the other hand, for both farms and manufacturers, understanding the economics of owning and operating this machine is crucial.

## 3.6. Comparative analysis

The empirical findings of the present investigation are juxtaposed with those of other scholars globally in Table 4. Laboratory tests were done to verify the theory's validity and the experiments' dependability. The suggested system offers a notable advantage over other systems discussed in the literature due to its quick identification time and a recognition rate of up to 100%. This system excels in accurately identifying sugar-cane stalk nodes, allowing farmers to produce sugarcane seeds at a minimal expense. Engaging in partnerships with manufacturers would facilitate large-scale production and the fine-tuning of systems, leading to enhanced automation and productivity in farming.

## 4. Conclusion and future works

The current study aimed to overcome the problems associated with the current fully automatic sugarcane cutting machines, where a pair of color sensors was used in the current study to detect the buds that had previously been hand-colored with a distinctive color and cut them automatically, and the sugarcane seed exit chute was provided with a sugarcane seed monitoring unit. The machine's performance was evaluated by measuring the damage index at sugarcane stalk diameters of (2.03, 2.72, 3.42, and 3.94 cm), two different types of the saw knives of the same diameter (7.0 in / 180 mm) (30 teeth, and 80 teeth), and five cutting times (1000, 1500, 2000, 2500, and 3000 ms), all tests were done at fixed cutting speed 12000 rpm. In

addition, the machine's performance was evaluated by conducting an economic analysis. The obtained results showed that the most damage index values were less than 0.00 for all cutting times and sugarcane stalk diameters under testing, while the DI values were equal zero (Partial damage) for sugarcane stalk diameter of 3.42 cm at cutting times of 2000 ms and 2500 ms. In addition, the economic analysis showed that the number of hectares that can be planted per day using the ASSCM is 0.45 and the total cost of sugarcane seeds per hectare is 70.865 USD.

## Recommendations

We suggest using a rotary saw knife with 80 teeth and a cutting time of 2000 ms to cut sugarcane stacks with an average diameter of 2.72 cm. This will result in higher performance and lower operating costs for the ASSCM.

## Future works

The results obtained from the present study represent a fundamental milestone towards further research and development of IoT and AI-powered technology for sugarcane harvesting. In particular, the utilization of IoT and AI can be extended to the automation of harvesting systems through the integration of robotics.

## Author Contributions

**Conceptualization:** Abdallah Elshawadfy Elwakeel, Kitmo.

**Funding acquisition:** Kitmo.

**Investigation:** Mohamed Elshahat Badawy, I. M. Elzein, Kitmo, Hany S. Hussein.

**Methodology:** Abdallah Elshawadfy Elwakeel, Kitmo, Manar A. Ourapi.

**Resources:** Mahmoud M. Hussein, Manar A. Ourapi.

**Software:** Abdallah Elshawadfy Elwakeel, Kitmo, Manar A. Ourapi.

**Supervision:** Loai S. Nasrat, Tamer M. El-Messery.

**Validation:** I. M. Elzein, Mohamed Metwally Mahmoud, Kitmo, Claude Nyambe.

**Writing – original draft:** Abdallah Elshawadfy Elwakeel, Kitmo, Manar A. Ourapi.

**Writing – review & editing:** Abdallah Elshawadfy Elwakeel, Loai S. Nasrat, I. M. Elzein, Mohamed Metwally Mahmoud, Mahmoud M. Hussein, Hany S. Hussein, Tamer M. El-Messery, Salah Elsayed.

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
