## [Decision Letter · Decision Letter 0]

8 May 2024

PONE-D-24-12311Advanced Design and Engineering Economical Evaluation of a Low-Cost Automatic Sugarcane Seed Cutting Machine Based RGB Color Sensor and IoT TechnologyPLOS ONE

Dear Dr. Kitmo .,

Thank you for submitting your manuscript to PLOS ONE. After careful consideration, we feel that it has merit but does not fully meet PLOS ONE’s publication criteria as it currently stands. Therefore, we invite you to submit a revised version of the manuscript that addresses the points raised during the review process.

We look forward to receiving your revised manuscript.

Kind regards,

Paulo Eduardo Teodoro, Dr.

Academic Editor

PLOS ONE

Journal Requirements:

4. We note that Figures 6,8,9 and 10 in your submission contain copyrighted images. All PLOS content is published under the Creative Commons Attribution License (CC BY 4.0), which means that the manuscript, images, and Supporting Information files will be freely available online, and any third party is permitted to access, download, copy, distribute, and use these materials in any way, even commercially, with proper attribution. For more information, see our copyright guidelines: http://journals.plos.org/plosone/s/licenses-and-copyright.

a. You may seek permission from the original copyright holder of Figures 6,8,9 and 10 to publish the content specifically under the CC BY 4.0 license. 

Reviewers' comments:

Reviewer's Responses to Questions

**Comments to the Author**

1. Is the manuscript technically sound, and do the data support the conclusions?

Reviewer #1: Yes

Reviewer #2: Partly

2. Has the statistical analysis been performed appropriately and rigorously? 

Reviewer #1: Yes

Reviewer #2: No

3. Have the authors made all data underlying the findings in their manuscript fully available?

Reviewer #1: Yes

Reviewer #2: No

4. Is the manuscript presented in an intelligible fashion and written in standard English?

Reviewer #1: Yes

Reviewer #2: Yes

5. Review Comments to the Author

Reviewer #1: The manuscript titled “Advanced Design and Engineering Economical Evaluation of a Low-Cost Automatic Sugarcane Seed Cutting Machine Based RGB Color Sensor and IoT Technology” is adequately structured, with an introduction that highlights the importance and the problem of study very well justified, although, the theoretical framework could be updated. The objectives are clear, demonstrated in the development of the study and in the results and conclusions obtained. The figures and tables presented are sufficient to characterize the experimental development, as well as to demonstrate the results obtained.

Some important points:

Justify the experimental choices and variables evaluated in the experiments (methodology). Although the economic analysis is important, it is limited to the region where the study was developed. Could this study be replicated to other sugarcane producing regions? In addition to the technology evaluated, the other evaluations seem to me to be more of a case study, applied to the region. Check the importance and meaning of the economic evaluation in the manuscript.

Expand the discussion with recent studies. The discussion is succinct and vague in relation to the good results obtained.

The conclusions are very extensive and vague. Present the conclusions punctually and on the proposed objectives, on the other hand, include part of the conclusions made in the discussion of the results.

Reviewer #2: Dear Editor,

After evaluating the manuscript titled "Advanced Design and Engineering Economical Evaluation of a Low-Cost Automatic Sugar-cane Seed Cutting Machine Based RGB Color Sensor and IoT Technology" my decision is: Major revision.

Overview: I acknowledge the importance of the topic covered in the manuscript and the development of intelligent mechanisms for sugar-cane seed cutting using sensors and IoT technology. However, the scientific contribution is limited. Therefore, if accepted for publication after the review process, I suggest it be considered as a technical article.

Title: The title is lengthy and confusing. It needs modification.

Abstract: The abstract should present results in a way that allows first-time readers to grasp the content easily. Some abbreviations used are not explained for readers unfamiliar with them. Furthermore, the abstract should include a discussion of the results and conclude with findings and recommendations.

Keywords: Avoid using terms already in the title.

Introduction: The introduction is inadequate and does not contribute effectively to understanding the work. The justification lacks depth in convincing the reader of the research's relevance. The problem is not clearly presented, making it difficult to understand the significance of this work. Specifically emphasize how developing an automatic sugar-cane seed cutting machine using color sensors and IoT can address identified challenges like improving efficiency and reducing operational costs. Additionally, the study's hypothesis is missing, hindering a comprehensive analysis of the manuscript's objectives. The manuscript's objectives need refinement, including specific aims. Furthermore, replace "[1]–[3]" and "[7]–[11]" with "[1–3]" and "[7–11]" throughout the manuscript. Update the citation "(Statista, 2018)" to [4] in line 55 with current data on sugar-cane production. Update reference [5] for the same reason in line 56.

Materials and Methods: The methodology is incomplete and needs detailed explanations of proposed procedures. More information is required for readers to replicate the experiment. Could the authors provide more details on specific procedures used for sensor calibration? For instance, what criteria were used to select reference colors, and what were the lighting conditions during calibration? Was there a validation process for sensor accuracy? How was sensor reliability in detecting and counting sugar-cane characteristics determined? A more detailed description of performance tests evaluating the machine's effectiveness would be beneficial. For example, how were cutting tests conducted for different stem diameters and working conditions? Were tests repeated for result consistency? Was reproducibility analyzed to ensure data consistency? What statistical methods were used to analyze test data? Statistical metrics evaluating accuracy and precision are necessary. Could the authors elaborate on these metrics?

Have specific considerations been made for operational risk assessment during machine development? How were quality controls implemented to ensure safe and consistent technology performance? Detailed documentation of research procedures, such as machine operation manuals, sensor calibration records, and test reports, should be provided as supplementary materials. Additional diagrams or illustrations clarifying methodological aspects, like detailed machine feeding, scanning, and cutting systems, would be helpful.

Results and Discussion: Results are well-presented but lack discussion and scientific justification. Weaknesses in the research are not addressed. For instance, manual procedures are still required despite machine use. Comparisons with relevant prior studies should also be included. This can contextualize results within the broader field and highlight unique research contributions. How do the economic analysis results demonstrate the proposed system's feasibility and potential benefits? Future research suggestions based on current findings should also be included.

Conclusions: Authors should address study objectives in this section.

Given the above, there are inconsistencies, and I recommend a "Major revision."

I encourage the authors to submit a revised version of the manuscript. If agreeable, I request responses addressing all points of disagreement or non-compliance.

Good luck.

Ad hoc Reviewer

6. PLOS authors have the option to publish the peer review history of their article (what does this mean?). If published, this will include your full peer review and any attached files.

Reviewer #1: No

Reviewer #2: No

---

## [Author Response · Author response to Decision Letter 0]

10 Jun 2024

***Technical response to the reviewers*** June 4th, 2024

Journal name: PLOS ONE 

No.: PONE-D-24-12311

OLD Title: “Advanced Design and Engineering Economical Evaluation of a Low-Cost Automatic Sugarcane Seed Cutting Machine Based RGB Color Sensor and IoT Technology”

NEW Title: “Advanced Design and Engi-Economical Evaluation of an Automatic Sugar-cane Seed Cutting Machine Based RGB Color Sensor” 

Abdallah Elshawadfy Elwakeel1, Loai S. Nasrat2, Mohamed Elshahat Badawy3, I. M. Elzein4, Mohamed Metwally Mahmoud5, Kitmo6&*, Mahmoud M. Hussein5,7, Hany S. Hussein2,8, Tamer M. El-Messery 9, Claude Nyambe 9, Salah Elsayed10 and Manar A. Ourapi1

1 Agricultural Engineering Department, Faculty of Agriculture and Natural Resources, Aswan University, Aswan 81528, Egypt; Abdallah_elshawadfy@agr.aswu.edu.eg, Manarourpi@gmail.com

2 Electrical Power Engineering Department, Faculty of Engineering, Aswan University, Aswan 81528, Egypt, loaisaad@yahoo.com

3 Agricultural Engineering Research Institute - Dokki – Giza 12611, Egypt, Mohamedelshahat@gmail.com

4Department of Electrical engineering, University of Doha for Science and Technology, Doha, Qatar

60101973@udst.edu.qa

5Electrical Engineering Department, Faculty of Energy Engineering, Aswan University, Aswan 81528, Egypt, Metwally_M@aswu.edu.eg

6&*University of Maroua, National Advanced School of Engineering of Maroua, Department of Renewable Energy, P.O. Box 46 Maroua, Cameroon, kitmobahn@gmail.com

7Department of Communications Technology Engineering, Technical College, Imam Ja’afar Al-Sadiq University, Baghdad, 10053, Iraq, mahmoud_hussein@aswu.edu.eg

8Electrical Engineering Department, College of Engineering, King Khalid University, Abha 62529, Saudi Arabia

hany.hussein@aswu.edu.eg

9International Research Centre “Biotechnologies of the Third Millennium”, Faculty of Biotechnologies (BioTech), ITMO University, St. Petersburg, 191002, Russia, telmesseri@itmo.ru; Clauderahl01@gmail.com

10Agricultural Engineering, Evaluation of Natural Resources Department, Environmental Studies and Research Institute, University of Sadat City, Sadat City 32897, Egypt; salah.emam@esri.usc.edu.eg

*Corresponding author: kitmobahn@gmail.com

Dear Editors and Reviewers

The authors are thankful to the learned Editors and Reviewers for their thoughtful and detailed comments to improve the quality of the manuscript. The authors have tried to address all the concerns, and the corrections are incorporated in the revised manuscript. The replies to the reviewer’s comments are provided below.

We hope that this revised version can meet the reviewer’s expectations and the standards for publication in PLOS ONE Journal.

The changes incorporated in the revised manuscript are highlighted in YELLOW.

Editor's Comments:

Our sincere thanks and appreciation to the Editors for recommending the submission of the revised manuscript with major revision. To improve the quality of the manuscript, the reviewer's queries are addressed, and their suggestions are incorporated into the revised manuscript. 

Reviewer Comments:

Reviewer 1:

Comments to the Authors:

Comment**: The manuscript titled “Advanced Design and Engineering Economical Evaluation of a Low-Cost Automatic Sugarcane Seed Cutting Machine Based RGB Color Sensor and IoT Technology” is adequately structured, with an introduction that highlights the importance and the problem of study very well justified, although, the theoretical framework could be updated. The objectives are clear, demonstrated in the development of the study and in the results and conclusions obtained. The figures and tables presented are sufficient to characterize the experimental development, as well as to demonstrate the results obtained.

 Authors response**: The authors are thankful to the honorable reviewer for the words of encouragement and trust in our work.

Comments1: Justify the experimental choices and variables evaluated in the experiments (methodology).

Response1: The authors are extremely thankful to the reviewer for this thoughtful point. We agree with you, and we corrected it, kindly check the updated paper (titles 2.4 & 2.5). 

Comment 2: Although the economic analysis is important, it is limited to the region where the study was developed. Could this study be replicated to other sugarcane producing regions? In addition to the technology evaluated, the other evaluations seem to me to be more of a case study, applied to the region. Check the importance and meaning of the economic evaluation in the manuscript.

Response 2: The authors are extremely thankful to the reviewer for this thoughtful point. We agree with you, and we added it, kindly check the updated paper (title 3.5 & Table 4). Where due to the lack of economic studies for the automatic sugarcane seed cutting systems, it is impossible to determine the economic efficiency of the manufactured machine in comparison to other systems worldwide. So, this study aimed to provide an economic analysis of the machine, which can be used as a reference point for evaluating and comparing the automatic sugarcane seed cutting systems in the future. On the other hand, for both farms and manufacturers, understanding the economics of owning and operating this machine is crucial.

Absolutely, this study can be replicated in any region, where all assumptions are shown in materials and methods and highlighted in yellow.

Comment 3: Expand the discussion with recent studies. The discussion is succinct and vague in relation to the good results obtained.

Response 3: The authors are extremely thankful to the reviewer for this thoughtful point. We agree with you, and we added it, kindly check the updated paper (Results and Discussion section).

Comment 4: The conclusions are very extensive and vague. Present the conclusions punctually and on the proposed objectives, on the other hand, include part of the conclusions made in the discussion of the results.

Response 4: The authors are extremely thankful to the reviewer for this thoughtful point. The authors completely agree with you, and we added it, kindly check the updated paper (Conclusions section).

Reviewer 2:

Comments to the Authors:

Comment**: After evaluating the manuscript titled "Advanced Design and Engineering Economical Evaluation of a Low-Cost Automatic Sugar-cane Seed Cutting Machine Based RGB Color Sensor and IoT Technology" my decision is: Major revision.

Overview: I acknowledge the importance of the topic covered in the manuscript and the development of intelligent mechanisms for sugar-cane seed cutting using sensors and IoT technology. However, the scientific contribution is limited. Therefore, if accepted for publication after the review process, I suggest it be considered as a technical article.

 Authors response**: The authors express gratitude to the reviewer for their encouragement and trust in our work. However, this paper is only a part of a larger project: 

1. https://doi.org/10.1371/journal.pone.0301294

2. https://aujes.journals.ekb.eg/article_345204_6670759135073095241fd79e475bc22f.pdf

Comment-1: Title: The title is lengthy and confusing. It needs modification.

Response-1: The authors are extremely thankful to the reviewer for this thoughtful point. We completely agree with you, and we added it, kindly check the updated paper (paper title).

Comment-2: The abstract should present results in a way that allows first-time readers to grasp the content easily. Some abbreviations used are not explained for readers unfamiliar with them. Furthermore, the abstract should include a discussion of the results and conclude with findings and recommendations.

Response-2: The authors are extremely thankful to the reviewer for this thoughtful point. The authors completely agree with you, and we added it, kindly check the updated paper (Abstract section).

Comment-3: Keywords: Avoid using terms already in the title.

Response-3: The authors are extremely thankful to the reviewer for this thoughtful point. We agree with you, so we developed the keywords. Kindly check the updated paper (keywords).

Comment-4: The introduction is inadequate and does not contribute effectively to understanding the work. The justification lacks depth in convincing the reader of the research's relevance. The problem is not clearly presented, making it difficult to understand the significance of this work. Specifically emphasize how developing an automatic sugar-cane seed cutting machine using color sensors and IoT can address identified challenges like improving efficiency and reducing operational costs. Additionally, the study's hypothesis is missing, hindering a comprehensive analysis of the manuscript's objectives. The manuscript's objectives need refinement, including specific aims. Furthermore, replace "[1]–[3]" and "[7]–[11]" with "[1–3]" and "[7–11]" throughout the manuscript. Update the citation "(Statista, 2018)" to [4] in line 55 with current data on sugar-cane production. Update reference [5] for the same reason in line 56.

Response-4: The authors are extremely thankful to the reviewer for this thoughtful point. We completely agree with you, and we adjusted and developed the introduction section, kindly check the updated paper (Introduction section).

Comment-5: The methodology is incomplete and needs detailed explanations of proposed procedures. More information is required for readers to replicate the experiment.

Response-5: The authors are extremely thankful to the reviewer for this thoughtful point. We completely agree with you, and we added it, kindly check the updated paper (Title 2.5).

Comment-6: Could the authors provide more details on specific procedures used for sensor calibration? For instance, what criteria were used to select reference colors, and what were the lighting conditions during calibration? Was there a validation process for sensor accuracy?

Response-6: The authors are extremely thankful to the reviewer for this thoughtful point. We completely agree with you, and we added it, kindly check the updated paper (Title 2.4 & subtitles 2.4.1. and 2.4.2.).

Comment-7: How was sensor reliability in detecting and counting sugar-cane characteristics determined.

Response-7: The authors are extremely thankful to the reviewer for this thoughtful point. But the Ultrasonic sensor was used only for counting the number of sugarcane seeds after the cutting process, the operating principle is shown in subtitles 2.2.4 and 2.3.2. & Figures 8 and 12. kindly check the updated paper (paper title).

Comment-8: A more detailed description of performance tests evaluating the machine's effectiveness would be beneficial. For example, how were cutting tests conducted for different stem diameters and working conditions? Were tests repeated for result consistency? Was reproducibility analyzed to ensure data consistency? What statistical methods were used to analyze test data? Statistical metrics evaluating accuracy and precision are necessary. Could the authors elaborate on these metrics?

Response-8: The authors are extremely thankful to the reviewer for this thoughtful point. 

We used the developed machine for cutting process, However, this paper is only a part of a larger project: 

1. https://doi.org/10.1371/journal.pone.0301294

2. https://aujes.journals.ekb.eg/article_345204_6670759135073095241fd79e475bc22f.pdf

Where the performance tests of the ASSCM were conducted under laboratory conditions in the Department of Agricultural Engineering, Aswan University, kindly check the updated paper (subtitle 2.5.).

All calibration tests were replicated at least three times, kindly check the updated paper (subtitle 2.4.1. & 2.4.2.).

We used many statistical analysis and programs for analysis the obtained data, kindly check the updated paper (subtitle 2.7. & 3.1.2. & 3.2.). 

Comment-9: How was sensor reliability in detecting and counting sugar-cane characteristics determined.

Response-9: This paper is a part of a larger project, and most of these points have been demonstrated in the current publications.

1. https://doi.org/10.1371/journal.pone.0301294

2. https://aujes.journals.ekb.eg/article_345204_6670759135073095241fd79e475bc22f.pdf

Comment-10: Results and Discussion: Results are well-presented but lack discussion and scientific justification. Weaknesses in the research are not addressed. For instance, manual procedures are still required despite machine use. Comparisons with relevant prior studies should also be included. This can contextualize results within the broader field and highlight unique research contributions. How do the economic analysis results demonstrate the proposed system's feasibility and potential benefits? Future research suggestions based on current findings should also be included.

Response-10: The authors are extremely thankful to the reviewer for this thoughtful point. We completely agree with you, and we adjusted and developed the Results and Discussion section, kindly check the updated paper (Results and Discussion section). Where the comparison of the ASSCM with other technologies has been demonstrated in subtitle 3.6. and table 4. In addition, Future research suggestions based on current findings has been demonstrated in future works in conclusion section. 

Comment-11: Conclusions: Authors should address study objectives in this section.

Response-11: The authors are extremely thankful to the reviewer for this thoughtful point. We completely agree with you, and we adjusted and developed the Conclusions section, kindly check the updated paper (Conclusions section).

The authors once again thank the learned Editors and Reviewers for their valuable comments for improving the quality of the manuscript.

---

## [Decision Letter · Decision Letter 1]

18 Jun 2024

Advanced Design and Engi-Economical Evaluation of an Automatic Sugarcane Seed Cutting Machine Based RGB Color Sensor

PONE-D-24-12311R1

Dear Dr. .,

We’re pleased to inform you that your manuscript has been judged scientifically suitable for publication and will be formally accepted for publication once it meets all outstanding technical requirements.

Kind regards,

Paulo Eduardo Teodoro, Dr.

Academic Editor

PLOS ONE

Additional Editor Comments (optional):

Reviewers' comments:

Reviewer's Responses to Questions

**Comments to the Author**

1. If the authors have adequately addressed your comments raised in a previous round of review and you feel that this manuscript is now acceptable for publication, you may indicate that here to bypass the “Comments to the Author” section, enter your conflict of interest statement in the “Confidential to Editor” section, and submit your "Accept" recommendation.

Reviewer #1: All comments have been addressed

Reviewer #2: All comments have been addressed

2. Is the manuscript technically sound, and do the data support the conclusions?

Reviewer #1: Yes

Reviewer #2: Partly

3. Has the statistical analysis been performed appropriately and rigorously? 

Reviewer #1: Yes

Reviewer #2: I Don't Know

4. Have the authors made all data underlying the findings in their manuscript fully available?

Reviewer #1: Yes

Reviewer #2: Yes

5. Is the manuscript presented in an intelligible fashion and written in standard English?

Reviewer #1: Yes

Reviewer #2: Yes

6. Review Comments to the Author

Reviewer #1: The authors revised the manuscript according to the comments submitted. The article has been adequately reviewed and is ready for acceptance in this form.

Reviewer #2: An improved version of the manuscript was presented. In it, the authors met all my considerations and I believe that the manuscript can now be accepted for publication.

7. PLOS authors have the option to publish the peer review history of their article (what does this mean?). If published, this will include your full peer review and any attached files.

Reviewer #1: No

Reviewer #2: No

---

## [Editor Report · Acceptance letter]

19 Aug 2024

PONE-D-24-12311R1 

PLOS ONE

Dear Dr. ., 

I'm pleased to inform you that your manuscript has been deemed suitable for publication in PLOS ONE. Congratulations! Your manuscript is now being handed over to our production team.

Kind regards, 

on behalf of

Professor Paulo Eduardo Teodoro 

Academic Editor

PLOS ONE